# Fatty Acid Amide Hydrolase Deficiency Is Associated with Deleterious Cardiac Effects after Myocardial Ischemia and Reperfusion in Mice

**DOI:** 10.3390/ijms232012690

**Published:** 2022-10-21

**Authors:** Sanela Rajlic, Luise Surmann, Pia Zimmermann, Christina Katharina Weisheit, Laura Bindila, Hendrik Treede, Markus Velten, Andreas Daiber, Georg Daniel Duerr

**Affiliations:** 1Department of Cardiothoracic and Vascular Surgery, University of Medicine Mainz, 55131 Mainz, Germany; 2Clinic for Oral and Maxillofacial Surgery, University Hospital Duesseldorf, 40225 Duesseldorf, Germany; 3CUROS Urology Center, 50668 Köln, Germany; 4Department of Anesthesiology and Operative Intensive Care Medicine, University Medical Center Bonn, 53127 Bonn, Germany; 5Institute of Physiological Chemistry, University of Medicine Mainz, 55128 Mainz, Germany; 6Center for Cardiology, Department of Cardiology, Molecular Cardiology, University Medical Center, 55131 Mainz, Germany; 7German Center for Cardiovascular Research (DZHK), Partner Site Rhine-Main, 55131 Mainz, Germany

**Keywords:** ischemic cardiomyopathy, fatty acid amide hydrolase, endocannabinoid receptor

## Abstract

Ischemic cardiomyopathy leads to inflammation and left ventricular (LV) dysfunction. Animal studies provided evidence for cardioprotective effects of the endocannabinoid system, including cardiomyocyte adaptation, inflammation, and remodeling. Cannabinoid type-2 receptor (CB2) deficiency led to increased apoptosis and infarctions with worsened LV function in ischemic cardiomyopathy. The aim of our study was to investigate a possible cardioprotective effect of endocannabinoid anandamide (AEA) after ischemia and reperfusion (I/R). Therefore, fatty acid amide hydrolase deficient (FAAH)^−/−^ mice were subjected to repetitive, daily, 15 min, left anterior descending artery (LAD) occlusion over 3 and 7 consecutive days. Interestingly, FAAH^−/−^ mice showed stigmata such as enhanced inflammation, cardiomyocyte loss, stronger remodeling, and persistent scar with deteriorated LV function compared to wild-type (WT) littermates. As endocannabinoids also activate PPAR-α (peroxisome proliferator-activated receptor), PPAR-α mediated effects of AEA were eliminated with PPAR-α antagonist GW6471 i.v. in FAAH^−/−^ mice. LV function was assessed using M-mode echocardiography. Immunohistochemical analysis revealed apoptosis, macrophage accumulation, collagen deposition, and remodeling. Hypertrophy was determined by cardiomyocyte area and heart weight/tibia length. Molecular analyses involved Taqman^®^ RT-qPCR and immune cells were analyzed with fluorescence-activated cell sorting (FACS). Most importantly, collagen deposition was reduced to WT levels when FAAH^−/−^ mice were treated with GW6471. Chemokine ligand-2 (CCL2) expression was significantly higher in FAAH^−/−^ mice compared to WT, followed by higher macrophage infiltration in infarcted areas, both being reversed by GW6471 treatment. Besides restoring antioxidative properties and contractile elements, PPAR-α antagonism also reversed hypertrophy and remodeling in FAAH^−/−^ mice. Finally, FAAH^−/−^-mice showed more substantial downregulation of PPAR-α compared to WT, suggesting a compensatory mechanism as endocannabinoids are also ligands for PPAR-α, and its activation causes lipotoxicity leading to cardiomyocyte apoptosis. Our study gives novel insights into the role of endocannabinoids acting via PPAR-α. We hypothesize that the increase in endocannabinoids may have partially detrimental effects on cardiomyocyte survival due to PPAR-α activation.

## 1. Introduction

Ischemic cardiomyopathy poses a significant threat to health worldwide [1]. Even though mortality rates from coronary artery diseases (CAD) have declined in the past 30 years [2], ischemic heart failure remains a fundamental problem because people with coronary artery diseases now live longer [3]. Myocardial hibernation is an adaptive condition of the heart at the early phase of CAD with decreased cardiac contractility due to oxygen shortage following interrupted coronary blood-flow intervals resulting from ischemia [4]. However, a general consensus is that myocardial hibernation may be reversed upon revascularization [5].

In this study, we used a murine model of brief, repetitive myocardial ischemia and reperfusion (I/R) with hallmarks of hibernating myocardium, e.g., reversibility of cardiac dysfunction. Repetitive, brief 15 min ischemia followed by 24 h of reperfusion over time mimics the pathomechanisms of CAD with an inflammatory response and left ventricular (LV) dysfunction in absence of myocardial infarction [6].

Endocannabinoids have played a tremendous role in researching ischemic cardiomyopathy in the past decade. In this regard, we could previously demonstrate the cardioprotective effects of the cannabinoid type-2 receptor (CB2) axis in the LV using murine models of ischemia and reperfusion [7,8]. The endocannabinoid system (ECS) is composed of multiple cannabinoid receptors, two of which are cannabinoid type-1 receptor (CB1) and CB2, being activated by endogenous ligands N-arachidonoylethanolamine (anandamide, AEA) and 2-arachidonoylglycerol (2-AG) [9,10]. With increasing research in this field, several other ligands and molecular targets with endocannabinoid-like effects have been identified, and as a result, the definition “endocannabinoid system” has gained complexity [11]. In addition, the ECS has become a potential pharmacological target of high interest. Concerning the I/R injury, recent studies have focused on the beneficial role of endocannabinoids protecting the ischemic heart via CB2 agonism in murine myocardial infarction [12,13,14]. Our group made a great effort to elucidate the role of the ECS—and more specifically the CB2—in ischemic cardiomyopathy, and our findings were consistent with the before-mentioned data, showing increased cardiomyocyte apoptosis and infarct size, followed by deteriorated LV function in CB2 deficient mice during repetitive myocardial I/R [7]. In order to confirm the potential cardioprotective role of CB2 activation, we sought for a method to activate CB2, keeping in mind that most agonists have partial antagonistic properties and their action was highly dose-dependent [12,15]. We, therefore, decided to take advantage of the ligands of the endogenous ECS [16].

### 1.1. Problem Definition

With this in mind, we have examined increased endogenous concentrations of AEA in FAAH^−/−^ (fatty acid amide hydrolase) mice undergoing repetitive I/R. The fatty acid amide hydrolase is the predominant catabolic enzyme of AEA, and its blockage has become a valuable target to amplify endocannabinoid signaling [17,18,19,20]. However, in contrast to the widely adopted theory, our data suggest that the beneficial cardiac effect of increased endocannabinoids in ischemic cardiomyopathy fails to appear. Instead, FAAH^−/−^ mice undergoing I/R experience a significant loss of cardiomyocytes, increased scar formation, and persistent LV dysfunction. On the other hand, wild-type (WT) mice have gained functional and morphological recovery. Thus, we have re-evaluated the problem by considering the evidence of endocannabinoids and their derivatives (AEA) activating Peroxisome proliferator-activated receptor (PPAR)-α in the ischemic heart [21,22,23]. In this regard, we were able to provide evidence of deleterious effects on the ischemic heart due to PPAR-α activation [24,25]. To secure contractility in altering situations, cardiomyocytes can metabolize different energy recourses, mainly fatty acids and carbohydrates. Generally, in the case of malperfusion and substrate shortage, the heart switches from fatty acid oxidation to glucose metabolism to meet the rising energy demand and prevent I/R injury [26,27]. Cardiac fatty acid-utilization pathways are partially controlled at the gene regulatory level. Previous studies have demonstrated an important role for the PPAR-α in the transcriptional control of genes involved in cardiac fatty acid-uptake and oxidation [28,29]. It is a key player in metabolic flexibility and nutrition regulation in the heart. PPARs belong to the nuclear hormone receptor family and include the alpha, beta, and gamma isotype [30]. PPAR activation mediates lipid metabolism, glucose utilization pathways, and inflammatory responses [31]. The alpha-isoform is intensely expressed in tissue with high beta-oxidation activity, such as the liver, skeletal muscle, and the heart [32]. In the energy metabolism of the heart, PPAR-α stimulates fatty acid uptake and beta-oxidation and reciprocally represses glucose uptake and oxidation [28,33,34,35,36]. Further, MHC-PPAR-α overexpressing hearts showed pathologies of diabetic cardiomyopathy, e.g., LV hypertrophy, enhanced gene expression of hypertrophy markers, and alteration in systolic LV dysfunction [28].

### 1.2. Study Motivation

Consequently, it was interesting to investigate PPAR-α inhibition in FAAH knockout mice in our model of ischemic cardiomyopathy undergoing brief, repetitive I/R. As a result, the PPAR-α-mediated effects of high AEA levels were eliminated with the selective PPAR-α antagonist GW6471 i.v. Furthermore, proof of principle was executed by blocking the effect of the PPAR-α agonist WY14.634 on PPAR-α downstream gene upregulation with antagonist GW6471 in WT mice.

### 1.3. Aims and Objective

The overall aim of this study was to work on clinical answers to fight the progression of ischemic cardiomyopathy at an early phase and gain insight into the metabolic interaction of enhanced ECS and the PPAR-α-mediated pathomechanisms.

## 2. Results

### 2.1. Fatty Acid Amide Hydrolase Deficient Mice Show Persistent Infiltration, Hypertrophy, and Loss of Function after I/R but Recover after PPAR-α Inhibition with GW6471

Repetitive brief I/R induced extensive cellular infiltration and vast intercellular space after 7d in the WT group, as already demonstrated [6] (Figure 1A). In contrast, FAAH^−/−^ mice revealed cellular infiltration in areas of cardiomyocyte loss (Figure 1B) determined with hematoxylin and eosin staining. Following 60d of recovery after 7d I/R, WT mice have gained functional and morphological recovery (Figure 1C), while the FAAH^−/−^ group did not show significant improvements, the areas of cardiomyocyte loss were filled by replacement fibrosis (Figure 1D). Progress in this group was also not significant when treated with DMSO (Figure 1E), while there was a noticeable recovery when treated with PPAR-α antagonist GW6471 (Figure 1F). The cardiomyocyte area was increased in the FAAH^−/−^ group compared to sham and WT group after 7d I/R (Figure 1G), but decreased after GW6471 treatment in FAAH^−/−^ group (Figure 1H). Heart weight/tibia length (HW/TL) as a parameter of myocardial hypertrophy was significantly increased after 7d I/R in the FAAH^−/−^ group (Figure 1I), indicating that FAAH deficiency elevates hypertrophy during I/R. However, this was significantly reduced after applying GW6471 in FAAH^−/−^ group (Figure 1J), implying that the hypertrophy developing mechanism results from PPAR-α activation. M-mode echocardiography showed decreased global LV function parameter fractional shortening (FS) in FAAH^−/−^ mice (Figure 1K) and reduced regional LV function parameter anterior wall thickening (AWT; Figure 1L). FS as well as AWT were restored in FAAH^−/−^ mice after GW6471 treatment (Figure 1M,N). In order to show the course of disease in these animals, we demonstrate data for each mouse longitudinally in Figure 1K.1–N.1.

### 2.2. Proof of Principle 

Firstly, genetic deletion of *FAAH* was demonstrated with lower mRNA levels of FAAH in this group of animals after 3 and 7d I/R (Figure 2A).

As previously mentioned, PPAR-α-mediated effects of high AEA levels were eliminated with the selective PPAR-α antagonist GW6471 i.v. Therefore, proof of principle was provided by blocking the effect of agonist WY14.634 on PPAR-α downstream genes upregulation, e.g., UCP3 (uncoupling protein 3, uncoupling electron transport chain; Figure 2B) and MCAD (Medium Chain Acyl CoA Dehydrogenase, mitochondrial β-oxidation; Figure 2C) by using antagonist GW6471 in WT mice after I/R. Also, UCP3 and MCAD-expression were evaluated in FAAH^−/−^ mice with DMSO (Dimethyl sulfoxide) and antagonist GW6471 treatment after I/R. Our data confirm the significant downregulation of UCP3 and MCAD expression in response to PPAR-α inhibition with GW6471 in FAAH^−/−^ mice and in WY14.634 treated WT mice after I/R. In contrast, we did not observe any changes when the WT group was treated only with GW6471, indicating that PPAR-α activation is specific for FAAH^−/−^ animals.

Further, FAAH deletion leads to lower degradation and therefore higher AEA levels compared to WT, as evaluated by LC-MS (liquid chromatography–mass spectrometry). This was not changed by PPAR-α antagonism via GW6471 (Figure 2D), confirming constantly elevated AEA levels in all FAAH groups.

FAAH^−/−^ mice show significant downregulation of PPAR-α gene expression (Figure 2E) after 3d I/R, suggesting a compensatory mechanism, as endocannabinoids are also ligands for PPAR-α. However, after GW6471 treatment, PPAR-α levels were significantly elevated in FAAH^−/−^ mice, and comparable to WT (Figure 2F). 

We measured APN (natriuretic Peptide A) gene expression in order to evaluate cardiac stress and found upregulation of it in the FAAH^−/^ group when compared with WT (Figure 2G) and downregulation after GW6471 treatment in FAAH knockout.

These data prove AEA action via PPAR-α, therefore allowing us to attribute the effects of elevated AEA levels in FAAH^−/−^ mice to PPAR-α agonism in our present study. In addition, it shows enhanced cardiac stress in FAAH^−/−^ mice, but normalization when after PPAR-α inhibition.

### 2.3. Fatty Acid Amide Hydrolase Deficiency Modulates Inflammation and Adaptation to Ischemia and Reperfusion

MAC-2 staining of representative LV sections after 3 and 7d I/R reveals macrophage infiltration predominantly located interstitially in WT mice, while FAAH^−/−^ mice show significantly higher macrophage density in areas of cardiomyocyte loss, being confirmed by cell count (Figure 3A). Notably, these effects in FAAH^−/−^ mice were reversed after PPAR-α antagonism with GW6471 (Figure 3B).

We observed significantly higher mRNA induction of pro-inflammatory macrophage chemoattractant protein CCL2 (chemokine ligand-2; Figure 3C) in FAAH^−/−^ mice after 3d I/R compared to sham and WT, but after GW6471 treatment, those effects were normalized (Figure 3D). Interestingly, the amount of total (Figure 3E) and dead (Figure 3F) immune cells was not changed between the groups. However, apoptosis of anti-inflammatory CD4^+^cells was increased in the FAAH^−/−^ group and decreased after using PPAR-α antagonist GW6471 (Figure 3G), and apoptosis of cytotoxic CD8^+^cells was adverse (Figure 3H).

The mRNA expression of antioxidative mediator heme oxygenase-1 (HMOX 1) was significantly less induced in FAAH^−/−^ after 3d I/R when compared to significant induction in WT (Figure 3I); This effect was reversed by treatment with GW6471 (Figure 3J).

Because hibernating myocardium is characterized by contractile dysfunction, we looked at myosin heavy-chain (MHC) expression. FAAH deletion caused a decrease in ATP saving MHC-ß isoform induction in response to I/R (Figure 3K). Nevertheless, there was no significant induction in FAAH^−/−^ mice treated with DMSO or GW6471 (Figure 3L).

These results suggest that PPAR-α-mediated cardioprotective antioxidative and energy-saving mechanisms fail in FAAH deficiency.

### 2.4. Fatty Acid Amide Hydrolase Deficiency Induces Hypertrophy and Increases Remodeling with High Myofibroblasts Accumulation and Collagen Deposition

Induction of macrophage differentiation marker OPN (osteopontin) was enhanced in FAAH^−/−^ mice compared to WT after 3d I/R (Figure 4A) and subordinated after GW6471 treatment (Figure 4B). Furthermore, significant upregulation of remodeling-related mediator tenascin-C (TNC) in WT after 3 and 7d I/R was abrogated in FAAH^−/−^ mice (Figure 4C), but restored upon GW6471 application (Figure 4D).

Representative LV sections stained for myofibroblast marker αSMAC (alpha Smooth Muscle Actin) in Figure 4 (panel E.1) show only a few interstitial myofibroblasts in WT hearts after 7d I/R. In contrast, numerous myofibroblasts were found in the ischemic myocardium of FAAH^−/−^ hearts (panel E.2). Those adverse effects mainly were absent when treated with GW6471, which is also objectivated by planimetry (panel E.3; Figure 4E,F).

The total collagen stained area was significantly higher in FAAH^−/−^ hearts after 3 and 7d I/R compared to WT and sham, but returned to WT levels after GW6471 treatment (Figure 4G,H). Representative sections stained for Picro Sirius red visualize these findings (panels G.1–G.3). In addition, while WT hearts almost completely restored collagen content to sham levels after 60d recovery from 7d I/R, FAAH^−/−^ hearts did not recover and showed persistently enhanced collagen density (Figure 4G).

Gene expression of metalloproteinases (MMPs) and tissue inhibitors of metalloproteinases (TIMPs) were measured with Taqman^®^ RT-qPCR, and all data are presented as ratio in the Figure 4I–K. In brief, the MMP8/TIMP1 ratio showed a decreasing tendency in all of the groups as well as the MMP13/TIMP1 ratio. MMP2/TIMP4 ratio was enhanced in FAAH^−/−^ hearts compared to sham and WT after 3 (tendency) 7d I/R (*p* < 0.05), but brought back to WT levels in FAAH^−/−^ mice after treatment with GW6471.

### 2.5. PPAR-α Antagonist Reduces Enhanced Apoptosis and Cardiomyocyte Loss in FAAH^−/−^ Mice to WT Levels

Cleaved caspase-3 (cCasp3) staining was performed to visualize apoptotic cardiomyocytes, and positive cardiomyocytes were detected by morphological means, as previously described [37], revealing significantly higher cardiomyocyte apoptosis in FAAH^−/−^ mice compared to WT after 3d I/R. Lower apoptosis was observed in FAAH^−/−^ mice after GW6471 treatment (Figure 5A.1–A.3). To corroborate our findings of apoptotic cardiomyocytes, we performed triple staining (Tunel + Troponin T + DAPI; 4′,6-diamidino-2-phenylindole) after 3d I/R in representative sections and confirmed a markedly higher amount of TUNEL positive cardiomyocyte nuclei in FAAH^−/−^ group compared to WT, as well as a highly reduced amount in FAAH^−/−^ mice after GW6471 treatment (panels Figure 5B.1–B.3).

## 3. Discussion

The aim of this study was to corroborate our previous findings that the ECS is involved in the development of ischemic cardiomyopathy: We previously showed that the absence of the CB2 receptor led to detrimental consequences for the ischemic heart, e.g., prolonged inflammation, increased remodeling, cardiomyocyte loss, replacement fibrosis, and reduced LV function [7].

Consequently, we hypothesized that CB2 activation led to ameliorated outcomes after I/R in our mouse model. Keeping in mind that most CB2 agonists have dose-dependent partial antagonistic properties [12,15,38], we subjected FAAH^−/−^ mice, providing ~15-fold increase in AEA levels [39], to repetitive, brief I/R. In this regard, it has been propagated that the ECS, through activation of the CB2 receptor, appears to be an important cardioprotective mechanism in isolated ischemic rat hearts [16].

However, in opposition to this widely adopted theory, our data show that the hypothesized beneficial cardiac effect of increased AEA in the ischemic murine heart fails to appear in our in vivo closed chest model of I/R. In contrast, FAAH^−/−^ mice showed enhanced inflammatory cell infiltration in areas of cardiomyocyte loss, followed by stronger remodeling with development of replacement fibrosis, scar formation, and persistently reduced LV function. In order to corroborate these findings, we evaluated cardiac stress by PPAR-α activation in FAAH^−/−^ mice by measuring ANP expression [40]; notably, it was upregulated in FAAH^−/−^ mice compared to WT in response to I/R, but PPAR-α antagonism with GW6471 reversed this effect.

This led to the assumption that our findings in FAAH^−/−^ mice might be related to ligand–receptor interaction other than AEA and CB2. With respect to this, AEA not only activates CB2, but has also shown to be a ligand for PPAR-α [41]. Hence, our data appear in a new light as we previously demonstrated that pharmacological activation of PPAR-α or its cardiomyocyte-specific overexpression was detrimental in ischemic cardiomyopathy [24,25]. In order to evaluate if these detrimental effects in FAAH^−/−^ mice are exerted via AEA–PPAR-α interaction, we applied the selective PPAR-α antagonist GW6471 to inhibit PPAR-α-mediated effects in the FAAH^−/−^ mice.

As proof of principle, we showed that the PPAR-α agonist WY14.634 induced upregulation of PPAR-α downstream genes UCP3 and MCAD [42] was reversed by GW6471 treatment in WT mice after I/R. Of note, no changes in gene regulation were observed in WT mice treated with GW6471. Importantly, we could also prove UCP3 and MCAD downregulation when FAAH^−/−^ mice were treated with PPAR-α antagonist GW6471, suggesting that PPAR-α activation is specific for FAAH^−/−^-animals, and that elevated AEA levels in FAAH^−/−^ mice indeed acted as ligands for PPAR-α. Moreover, we demonstrated elevated AEA levels in FAAH-deficiency irrespective of treatment with GW6471 or its vehicle DMSO, indicating that PPAR-α antagonism did not affect production or degradation of AEA. Interestingly, we noticed significant PPAR-α downregulation in FAAH^−/−^ mice suggesting a compensatory mechanism in order to attenuate lipotoxicity and cardiomyocyte loss possibly caused by PPAR-α activation [24,25]. Of note, GW6471 treatment restored PPAR-α levels in FAAH^−/−^ mice.

Repetitive, brief I/R caused extensive cellular infiltration and vast intercellular space in the WT group, as previously shown [6]. In contrast, the FAAH^−/−^ group showed cardiomyocyte loss, enhanced inflammation with inflammatory cell accumulation, intensified extracellular matrix (ECM) remodeling, and subsequent replacement fibrosis. Moreover, cardiomyocyte loss led to increased compensatory cardiomyocyte hypertrophy, but also deteriorated LV function compared to WT. This was also described in the MHC-PPAR-α overexpressing mice [6]. These devastating findings in FAAH^−/−^ mice were mostly reversed after PPAR-α antagonism with GW6471, supporting our hypothesis that AEA activates PPAR-α. Most importantly, 60d after I/R, recovery of LV function was absent in FAAH^−/−^ mice compared to WT, but restored by PPAR-α antagonism with GW6471.

With respect to the preceding inflammation, FAAH^−/−^ mice showed significantly increased CCL2 induction followed by higher macrophage density in areas of cardiomyocyte loss. These effects in FAAH^−/−^ mice were reduced after GW6471 treatment, suggesting that detrimental effects on cardiomyocyte survival and inflammation are indeed due to PPAR-α activation. Further, a relation between increased CCL2 and worse LV function in FAAH^−/−^ mice is supported by several clinical studies that have demonstrated a correlation between increased CCL2 levels and worse outcomes after cardiac injury [43,44,45]. Interestingly, while the amount of total and dead immune cells was not changed, the amount of dead, Annexin V positive CD4^+^cells was increased, but Annexin V positive CD8^+^cells decreased in FAAH^−/−^ mice, indicating an enhanced immune response in FAAH deficiency. This was reversed by GW6471 treatment, corroborating the pro-inflammatory properties of PPAR-α activation via AEA. Taken together, our data shed new light on the ECS and PPAR-α being involved in the modulation of the inflammatory response in the development of ischemic cardiomyopathy.

The recurrent hypoxia in the I/R model leads to an increased formation of cytotoxic ROS [6,46], which in high concentrations activate antioxidant enzymes such as HMOX 1 [47]. In response to I/R, FAAH^−/−^ mice showed lower HMOX 1 induction than WT, being reversed by GW6471 treatment. This restricted antioxidant capacity in FAAH^−/−^ animals after I/R supports the finding of cardiomyocyte loss and subsequently worsened myocardial function, as previously described [7,8,48]. This is in line with former studies showing that increased PPAR-α activation is associated with decreased HMOX 1 expression and worsened LV dysfunction [24,25].

Considering that ischemic cardiomyopathy is characterized by contractile dysfunction, we looked at myosin iso-gene expression. MHC-β reveals decreased contractile velocity but conserves more ATP per contraction than MHC-α [49]. Under physiological conditions, rodent hearts consist of <10% MHC-β [50]. MHC-β increase is reported in pressure overload, hypoxia and I/R [7,51] is believed to partly contribute to transient contractile dysfunction in order to preserve cardiac integrity [52,53]. Here, FAAH deletion caused absent MHC-β induction in response to I/R. This might be interpreted as missing cardiomyocyte adaptation to I/R injury in FAAH deficiency, and could also contribute to cardiomyocyte damage and absent recovery of LV function 60d after I/R. Hence, our data imply that cardioprotective properties such as antioxidant and energy-saving mechanisms fail in FAAH deficiency due to PPAR-α-mediated mechanisms.

I/R led to a stronger increase in macrophage differentiation and remodeling marker OPN in FAAH^−/−^ mice compared to WT, declining after GW6471 treatment. This suggests prolonged macrophage activity in FAAH deficiency when PPAR-α is activated and implies prolonged myocardial remodeling [54]. In this regard, we observed that upregulation of remodeling-related marker TNC, present in WT hearts, was significantly reduced in FAAH^−/−^ mice but restored after GW6471 application in the latter. This is in line with the previous finding of TNC downregulation in MHC-PPAR-α overexpressing mice [25]. It therefore supports the hypothesis of prolonged myocardial remodeling based on a weak expression of TNC. Restoration of OPN as well as TNC expression after GW6471 application in FAAH^−/−^ mice might explain the return to physiological ECM remodeling comparable to WT.

In our mouse model of repetitive I/R, several stimuli promoted the differentiation of cardiac fibroblasts to myofibroblasts [55,56]. Stronger remodeling with high αSMAC positive myofibroblasts’ accumulation in the ischemic heart was observed in FAAH^−/−^ mice and was diminished by PPAR-α antagonism using GW6471. Subsequently, collagen deposition was significantly elevated in FAAH^−/−^ hearts and located in areas of cardiomyocyte loss, e.g., replacement fibrosis, but returned to WT levels by GW6471 treatment. These findings are in accordance with findings from MHC-PPAR-α overexpressing mice [25], showing stronger myofibroblasts’ accumulation and fibrosis. Hence, our data show that fibrosis followed myofibroblast accumulation and returned to WT levels after GW6471 treatment in FAAH deficiency, thus corroborating our hypothesis that AEA also activates PPAR-α.

The attenuation of remodeling after GW6471 treatment is further underlined by modulation in MMP and TIMP expression. After myocardial infarction, it is a generally accepted pathway that the homeostatic balance between extracellular matrix degrading MMPs and their inhibitors (TIMPs) is disrupted, leading to early extracellular matrix damage and LV remodeling [57]. Further, imbalance in the MMP/TIMP system was significantly correlated to the development of ventricular dilatation [58]. MMP upregulation is involved in collagen degradation and LV dilation leading to dysfunction [59]. MMPs, including MMP13 and MMP8, are involved in extracellular matrix turnover in myocardial damage [60]. Previous studies showed that MMP inhibition preserved LV function in animal heart-failure models [61,62]. TIMP1 is the major inhibitor of MMP3, 9 and MMP13 [63]. Concomitantly, we show strongly reduced MMP13/TIMP1 as well as MMP8/TIMP1 ratios, indicative of compensatory inhibition of LV matrix turnover in order to preserve function.

TIMP2 inhibits MMP2 more effectively than the other TIMPs [63]. The MMP2/TIMP4 ratio was detected as a marker of myocardial dysfunction and as a negative predictor for survival in idiopathic pulmonary hypertension [48,64]. With that in line, our data show a significantly enhanced MMP2/TIMP4 ratio only in FAAH^−/−^ mice after repetitive I/R, and it is accompanied by deteriorated LV function. We therefore speculate that antagonism of PPAR-α leads to a differential regulation of MMPs/TIMPs, contributing to a modulated inflammatory response and modulation of ECM remodeling.

Most importantly, while hearts from the WT group show a complete functional and morphological recovery 60d after discontinuation of I/R, FAAH^−/−^ mice did not recover, demonstrating persistent scarring and LV dysfunction. These observations show that FAAH deficiency indeed induces hypertrophy and enhanced remodeling with high myofibroblasts’ accumulation, MMP/TIMP dysregulation and subsequent collagen deposition in areas of cardiomyocyte loss, leading to deteriorated LV function, as summarized in the schematic overview in Figure 6. However, we also demonstrate that this can be reversed to WT levels with PPAR-α antagonism. This underlines our hypothesis that elevated AEA levels also act via other receptors than CB2, e.g., PPAR-α.

Taken together, our study demonstrates that all mechanisms leading to restoration of the pathophysiological parameters in FAAH^−/−^ mice are achieved by blocking AEA action on PPAR-α. In summary, we show that missing cardioprotective effects in animals providing elevated AEA levels are caused by the lack of AEA selectivity to the CB2 receptor. In this context, one has to keep in mind that endocannabinoids such as AEA are also fatty acids and therefore affect receptors involved in fatty acid metabolism, such as the PPAR-α. In this light, the molecular, morphological, and functional effects found in FAAH^−/−^ mice are at least in part attributable to AEA action via PPAR-α.

### Limitation of the Study

In the present study, we could indicate that the elevated AEA levels are not a potential way to provide cardioprotection via CB2 in murine I/R, but that other measures have to be taken in order to provide cardioprotection via CB2. Instead of using elevated endocannabinoid concentrations, the best chance for proving the beneficial effects of CB2 activation in ischemic heart disease would possibly be to specifically activate CB2. In case this will still be out of reach due to the lack of selectivity for CB2, one could still take advantage of knockout mice for other endocannabinoid degrading enzymes, e.g., monoacylglycerol lipase (MAGL) and diacylglycerol lipase (DAGL).

A study of Mukhopadhyay et al. described comparable findings with respect to deleterious cardiac effects in FAAH deficiency. In contrast to our study, myocardial injury was induced with chemo-therapeutic drug, doxorubicin (DOX), that is known for its cardiotoxicity mediated by increased reactive oxygen and nitrogen species generation [18]. In their work, the authors aimed at the AEA-CB1 ligand–receptor interaction and showed that CB1 inhibition attenuated the DOX-mediated effects in FAAH knockout mice, revealing the minor selectivity of AEA on PPAR and endocannabinoid receptors.

In this regard, a recent article revealed that GW6471 also exhibited affinity for PPAR-γ with LanthaScreen™ TR-FRET binding assay [65]. We used GW6471 as described by Kapoor et al., as they showed highly selective PPAR-α antagonism in vivo [66]. However, one could still argue that future experiments on that topic should use a more specific PPAR-α antagonist. 

As for evaluation of cardiomyocyte apoptosis, cryosections of cardiac samples would have provided better options for immunofluorescent multiple antibody staining, which is restricted in paraffin-embedded sections due to loss of antigenicity or epitope instability [67].

Finally, a methodological limitation of our study is the absent protein analysis of inflammatory reaction; this was due to a shortage of mice.

## 4. Materials and Methods

### 4.1. Study Animals

Experimental procedures were carried out on 10–12 week old mice and were in accordance with the animal protocol approved by the local governmental authorities (84-02.04.2013.A330) and according to the European Union Directive 2010/63/EU for animal research. As previously described, the FAAH gene was isolated from a 129SvJ genomic library [68]. A PGK-Neo cassette was added between *Eco*RI and *Eco*RV sites placed 2.3 kb aside, replacing the first FAAH exon (encoding amino acids 1–65) and ≈2 kb of upstream sequence. Homologous recombinant 129SvJ embryonic stem cell clones were detected by Southern analysis, and two of those clones were used to generate chimeric mice on a C57BL/6 background. Chimeras from both clones gave germline transmission of the mutated gene. All mice used in this study were second or third generation offspring from intercrosses of 129SvJ-C57BL/6 FAAH^+/−^ mice. 

### 4.2. Brief Repetitive I/R Protocol

In the initial surgical procedure, the left descending coronary artery (LAD) ligation was performed as previously described [6]. After recovering for 7 days, the mice underwent daily 15 min occlusion of the LAD for 3 or 7 repetitive days with subsequent reperfusion. Before organ harvesting, cardiac function was assessed using M-mode echocardiography as described before. Hereafter, hearts were excised and further prepared according to the needed examination method [54].

### 4.3. Assessment of Left Ventricular Function—M-Mode Echocardiography

M-mode echocardiography with a two dimensional guided 15-MHz probe (HDI-5000; ATL Philips, Oceanside, CA, USA) was performed in WT and FAAH^−/−^ mice with DMSO and GW6471 treatment as previously published [25,54]. The echocardiographic results are longitudinal experiments. After initial surgery for implantation of the LAD-ligature (but NOT yet LAD occlusion), mice were allowed to recover for a period of 7 days in order to avoid measuring inflammation due to thoracotomy. In these mice, echocardiography was performed 7 days after initial surgery, but before the first day of LAD occlusion. After 7 days recovery from thoracotomy, the first I/R episode was performed by exhibiting the ligatures being stored subcutaneously. After 7 consecutive days, the second echocardiography was performed in the same mice, and was repeated after 60 days recovery from I/R. Only mice that survived all time points and underwent all echocardiographic evaluations were included in this experiment in order to connect individual measurements, showing the course of disease in these animals.

### 4.4. Treatment with PPAR-α Agonist WY14.634 in WT and PPAR-α Agonism with GW6471

To study the role of PPAR-α inhibition, PPAR-α-mediated effects of high AEA levels in FAAH^−/−^ mice were blocked with selective PPAR-α antagonist GW6471 (Tocris Bioscience, Bristol, UK). To show PPAR-α specificity for FAAH^−/−^ animals, we treated WT mice with GW6471 and did not observe any changes compared to WT, indicating FAAH^−/−^ specificity of GW6471 treatment in the presented study. Proof of principle was executed by blocking the effect of PPAR-α agonist WY14.634 (0.3 mg/kg in 10% DMSO with 0.9% saline, i.v. injection in the jugular vein 30 min before I/R) [69] on PPAR-α downstream gene upregulation with antagonist GW6471 in WT as previously described in two different in vivo studies in mice [66,70]. Intravenous injection of 100µL of either GW6471 (1 mg/kg i.v.) or its vehicle (10% DMSO with 0.9% saline) was performed daily 30 min before LAD occlusion.

### 4.5. Hypertrophy

Hearts were dissected and rinsed in ice-cold cardioplegic solution after removing the atria. Before weight measurement, the hearts were dried on a paper towel for five seconds. To analyze cardiac hypertrophy, heart weight/tibia length ratio was calculated. 

### 4.6. Histology

After paraffinization and microtome cutting, the hearts were stained for basic histopathology with hematoxylin and eosin (HE) and Picro Sirius red staining (SR) as already published [6]. The analysis of collagen distribution was completed by evaluating mosaic pictures of left ventricles at 100× magnification [48]. The total collagen area in the LV were related to the complete LV wall and presented as a percentage. In addition, planimetric evaluation of cardiomyocyte cross-sectional area in collagen-stained sections was performed in 400× magnification to assess cardiomyocyte hypertrophy [37].

### 4.7. Immunohistochemistry and Immunofluorescence 

For immunohistochemistry staining of paraffin-embedded tissue sections, Elite ABC kits and diaminobenzidine (DAB; AXXORA, Lörrach, Germany) were used. Cell density was described as cells per square millimeter, as previously reported [6,7]. For mouse-derived antibodies, a mouse-on-mouse immunodetection kit (AXXORA) was used. On paraffin-embedded slides, the following primary antibodies were used: macrophages, MAC-2 rat anti-mouse antibody (clone 3/38; Cedarlane, ON, Canada); myofibroblasts, α-smooth muscle actin (αSMAC) mouse anti-mouse monoclonal antibody (clone 1A4; Sigma, Taufkirchen, Germany) [7,8]. For detection of apoptotic cardiomyocytes, cleaved ASP 175 Caspase-3 rabbit anti-mouse polyclonal antibody (Cell Signaling, Danvers, MA, USA) was used, and a biotinylated secondary antibody was used (DAKO, Hamburg, Germany) for the latter. These apoptotic cardiomyocytes were detected by morphological means, as cCasp3 is a cytoplasmic staining, allowing to determine cardiomyocytes by morphology [37]. For MAC-2 and cCasp3, counterstaining was performed with Quick hemalaun kit (Vector, Burlingame, CA, USA), and for αSMAC with eosin as described [25]. 

### 4.8. Triple-Immunofluorescence Staining for Cardiomyocytes(Troponin T), TUNEL, and DAPI

In order to determine specific apoptotic cardiomyocytes, we performed double immunofluorescence staining with TUNEL staining (In Situ Cell Death Detection Kit, Fluorescein; Roche Applied Science, Penzberg, Germany) and cardiac marker (troponin, ab33589; Abcam, Cambridge, MA, USA) as previously described [71], and DAPI was used for nuclei staining (Thermo scientific, Rockford, IL, USA). Slides were scanned with the Leica TCS SP8, which is an inverse confocal fluorescence microscope based on an DMi8 CEL advanced stand (Leica Microsystems, Wetzlar, Germany) and images were taken using LAS X software (Version 10) 40× magnification.

### 4.9. Flow Cytometry Analysis

Single-cell suspensions from *n* = 5–6 hearts/group and time points were generated as previously described [72]. We stained for immune cells CD45 (30-F11) and adaptive immune cells using various antibodies from Thermo Fisher and BioLegend (San Diego, CA, USA) B220/CD45R (RA3-6B2), CD4 (RM4-5), CD8a (53-6.7). Annexin V staining for detecting apoptotic cells was performed according to the manufacturer’s protocol (Cat. No. 88-8005-72; Thermo Fisher Scientific). Live/dead staining was performed using Hoecht 33,342 solution (Thermo Fisher). For flow cytometry analysis, FACS-Canto II, LSR II, and Fortessa (BD Bioscience, Heidelberg, Germany) were used, and the data were analyzed with Flow-Jo software v10 (BD, Franklin Lakes, NJ). The gating strategy for flow cytometry analyses was performed as illustrated for myocardium (Appendix A).

### 4.10. Endocannabinoid Measurements by LC-MS/MS

Quantitative profiling of endocannabinoids was performed with an LC-MS/MS system (Agilent 1200 LC system; 5500 QTrap; AB SCIEX, Darmstadt, Germany). The concentration of anandamide was quantified and further equalized to the initial amount of protein. 

### 4.11. Gene Expression Analysis 

The mRNA expression was calculated with Taqman^®^ real-time quantitative RT-qPCR and FAM-TAMRA linked customized primers in an ABI Prism 7900HT Sequence Detection System and SDS 2.4 software (Applied Biosystems/Life Technologies, Karlsruhe, Germany) [73]. The mRNA levels were correlated with sham and GAPDH applying the comparative ΔΔCt-method [37].

### 4.12. Statistics

The results were tested for normal distribution and expressed as mean ± SEM. Comparison between the groups was performed by 1way ANOVA and Newman–Keuls post hoc testing (PRISM 9.3.1; GraphPad, La Jolla, CA, USA). Where appropriate, we applied Mann–Whitney U-Test (Figure 2B,C). Longitudinal echocardiographic measurements were analyzed by two-way ANOVA and Tukey’s post-hoc test (Figure 1K–N). Statistical significance was considered *p* < 0.05.

## Figures and Tables

**Figure 1 ijms-23-12690-f001:**
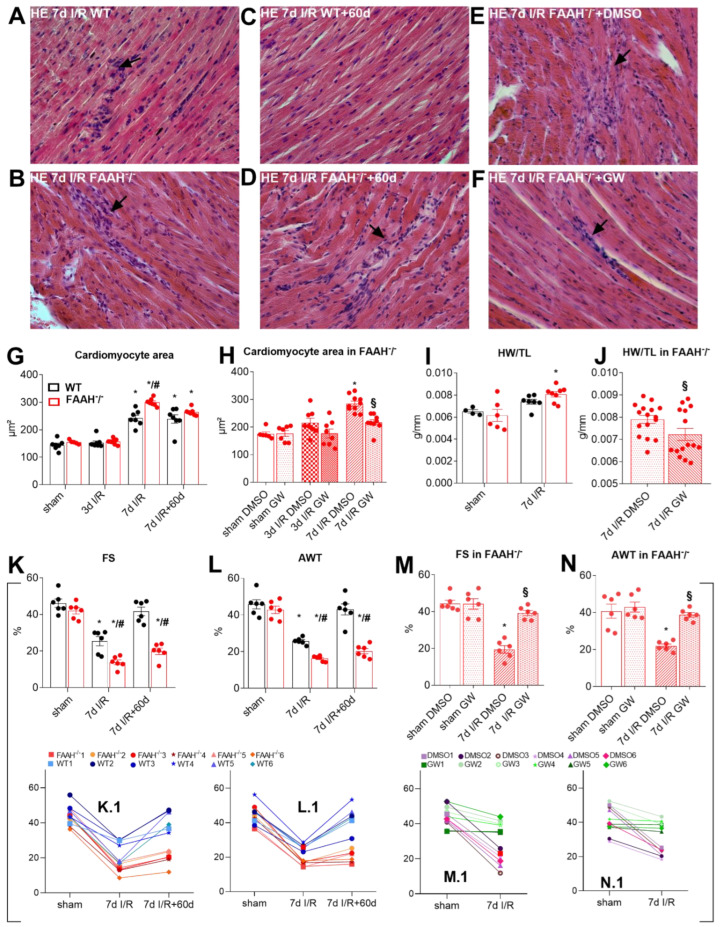
FAAH^−/−^ mice show persistent cellular infiltration, cardiomyocyte hypertrophy, and loss of LV function after I/R, but recovery after PPAR-α inhibition with GW6471. Repetitive, brief I/R induced interstitial cellular infiltration (arrows) in (**A**) WT and in areas of cardiomyocyte loss (**B**) in FAAH^−/−^ mice after 7d I/R as observed with hematoxylin and eosin staining. Panel (**C**) Shows recovery in WT after 60d, but persistent cellular infiltration in (**D**) FAAH^−/−^ mice. (**E**,**F**) Recovery of FAAH^−/−^ group after PPAR-α antagonism with GW6471 when compared to DMSO. (**G**) Increased cardiomyocyte area in FAAH^−/−^ after 7d I/R and (**H**) decrease after GW6471 treatment. (**I**) HW/TL ratio showing increased hypertrophy in FAAH^−/−^ group after 7d I/R, (**J**) normalizing after GW6471 treatment. Persistently reduced global and anterior LV function shown by (**K**) FS and (**L**) AWT in FAAH^−/−^ was restored after GW6471 treatment (**M**,**N**). (**K.1**–**N.1**) show individual mouse data over time connected through different time points, allowing comparison of pre- and post-intervention values longitudinally. Magnification in (**A**–**F**): 400×. AWT, anterior wall thickening; DMSO, Dimethyl sulfoxide; FAAH, fatty acid amide hydrolase; FS, fractional shortening; HE, hematoxylin eosin; HW/TL, heart weight/tibia length; WT, wild-type; *, *p* < 0.05 vs. respective sham; #, *p* < 0.05 FAAH^−/−^ vs. WT at the respective time point; §, *p* < 0.05 FAAH^−/−^+GW vs. FAAH^−/−^+DMSO at the respective time point.

**Figure 2 ijms-23-12690-f002:**
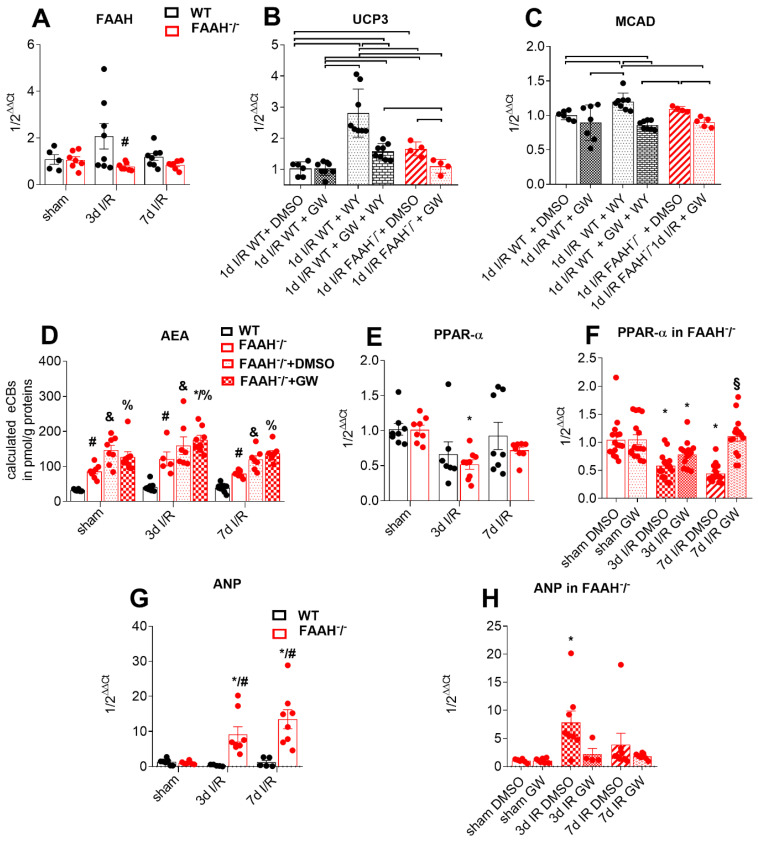
Proof of principle. (**A**) Lower FAAH mRNA levels in FAAH^−/−^ group. PPAR-α downstream gene UCP3 and MACD activation with agonist WY14.634 and its inhibition with antagonist GW6471 are shown in (**B**,**C**). (**D**) LC-MS reveals increased AEA levels in the FAAH^−/−^ group compared to WT, remaining unchanged by PPAR-α antagonist GW6471 and DMSO. (**E**) FAAH^−/−^ mice show more substantial downregulation of PPAR-α mRNA levels, and (**F**) those levels were restored by GW6471. (**G**) ANP is upregulated in FAAH^−/−^ group compared to WT, being reduced (**H**) after GW treatment. RT-qPCR using Taqman^®^, expression is related to sham and GAPDH using comparative ∆∆Ct method. AEA, Anandamide; ANP, natriuretic Peptide A; DMSO, Dimethyl sulfoxide; FAAH, fatty acid amide hydrolase; MCAD, Medium Chain Acyl CoA Dehydrogenase; PPAR-α, Peroxisome proliferator-activated receptor-alpha; UCP3, uncoupling protein 3; WT, wild-type; *, *p* < 0.05 vs. respective sham; #, *p* < 0.05 FAAH^−/−^ vs. WT at the same time point; §, *p* < 0.05 FAAH^−/−^+GW vs. FAAH^−/−^+DMSO at the respective time point; &, *p* < 0.05 FAAH^−/−^+DMSO vs. WT at the respective time point; %, *p* < 0.05 FAAH^−/−^+GW vs. WT at the respective time point.

**Figure 3 ijms-23-12690-f003:**
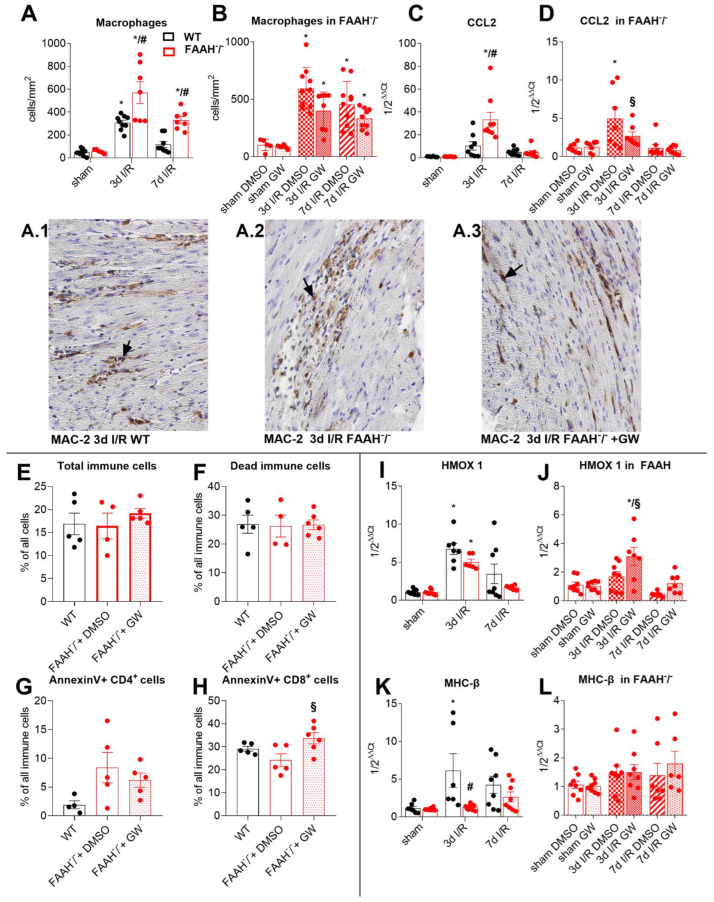
Fatty acid amide hydrolase deficiency modulates inflammation and adaptation. (**A**) Increased macrophage accumulation in the FAAH^−/−^ group compared to WT was (**B**) restored with GW6471 treatment (representative pictures in (**A.1**–**A.3**); macrophages accumulation is indicated with arrows). (**C**) Significantly increased CCL2 expression was observed in FAAH^−/−^ group, but (**D**) decreased after GW6471. (**E**) Amount of total immune cells and (**F**) dead immune cells was not changed. (**G**) Apoptosis of CD4^+^cells was increased in FAAH^–/−^ group, and slightly decreased after GW6471 application. (**H**) Contrarily, apoptosis of CD8^+^cells was reduced in FAAH^−/−^ group, but increased again after GW6471 treatment. (**I**) Levels of HMOX 1 were significantly reduced in FAAH^−/−^ group (**J**) but increased after GW6471 application. (**K**) Energetically efficient myosin heavy-chain isoform MHC-β was suppressed in FAAH^−/−^ mice, (**L**) but reversed after GW6471 treatment. RT-qPCR using Taqman^®^, expression is related to sham and GAPDH using comparative ∆∆Ct method. Magnification in (**A.1**–**A.3**): 400×. MHC-β, beta-myosin heavy chain; CCL2, chemokine ligand-2; DMSO, Dimethyl sulfoxide; FAAH, fatty acid amide hydrolase; HMOX 1, heme oxygenase 1 gene; IL-10, Interleukin-10; WT, wild-type. *, *p* < 0.05 vs. respective sham; #, *p* < 0.05 FAAH^−/−^ vs. WT at respective time point; §, *p* < 0.05 FAAH^−/−^+GW vs. FAAH^−/−^+DMSO at the respective time point.

**Figure 4 ijms-23-12690-f004:**
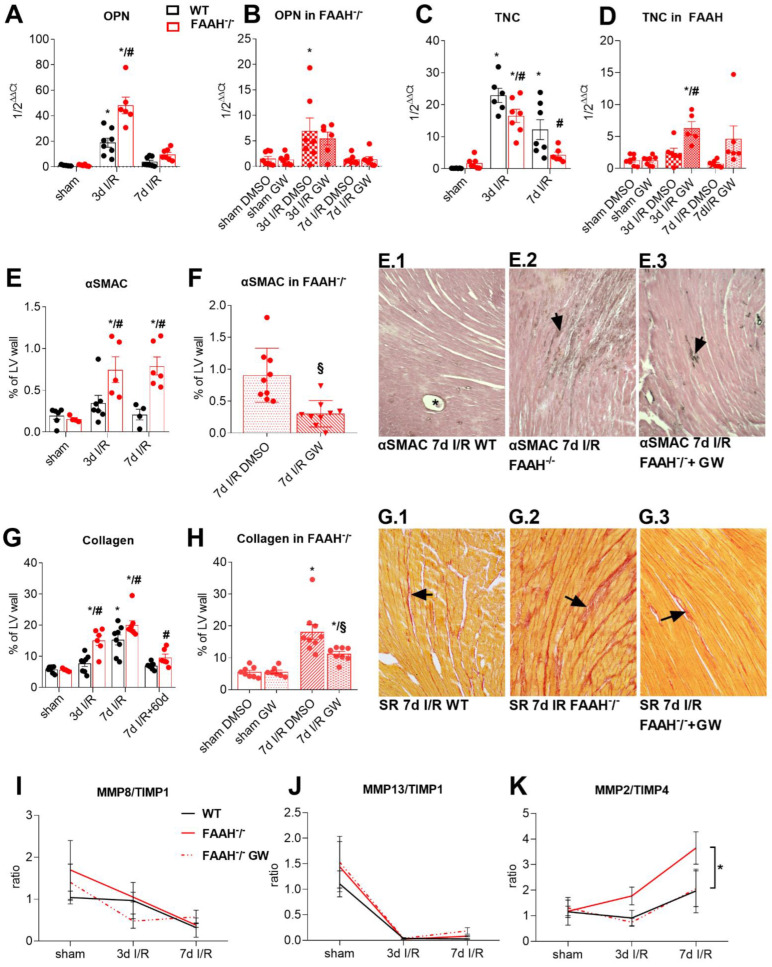
Fatty acid amide hydrolase deficiency induces stronger extracellular matrix remodeling with high myofibroblasts accumulation and collagen deposition. (**A**) Macrophage differentiation marker OPN was increased in FAAH^−/−^ group and (**B**) normalized with GW6471. (**C**) The mRNA expression of remodeling-related mediator tenascin-C was significantly lower in FAAH^−/−^ group after I/R compared to WT, and (**D**) restored after GW6471 treatment. (**E**) Intensified remodeling with high myofibroblasts accumulation observed in FAAH^−/−^ mice (**F**) was diminished using GW6471, as also shown in representative αSMAC staining in panels (**E.1**–**E.3)**, arrows show αSMAC positive myofibroblasts; asterisk in E1: small arteriole with αSMAC positive tunica muscularis. (**G**) Collagen deposition was increased in FAAH^−/−^ group after I/R and (**H**) reduced when FAAH^−/−^ mice were treated with GW6471. Representative Picro Sirius red staining in (**G.1**–**G.3**), arrows indicate collagen deposition. (**I**) MMP8/TIMP1 ratio shows decreasing tendency (**J**) together with MMP13/TIMP1 ratio, while (**K**) MMP2/TIM4 ratio shows increasing tendency. RT-qPCR using Taqman^®^, expression is related to sham and GAPDH using comparative ∆∆Ct method. Magnification in (**E.1**–**E.3**,**G.1**–**G.3**): 200×. αSMAC, alpha Smooth Muscle Actin; DMSO, Dimethyl sulfoxide; FAAH, fatty acid amide hydrolase; MMP2, matrix metalloproteinase-2; OPN, osteopontin; TNC, tenascin-C; WT, wild-type. *, *p* < 0.05 vs. respective sham; #, *p* < 0.05 FAAH^−/−^ vs. WT at respective time point; §, FAAH^−/−^+GW vs. FAAH^−/−^+DMSO at the respective time point.; Panel (**K**) *, *p* < 0.05 for 7d I/R WT vs. 7d I/R FAAH^−/−^.

**Figure 5 ijms-23-12690-f005:**
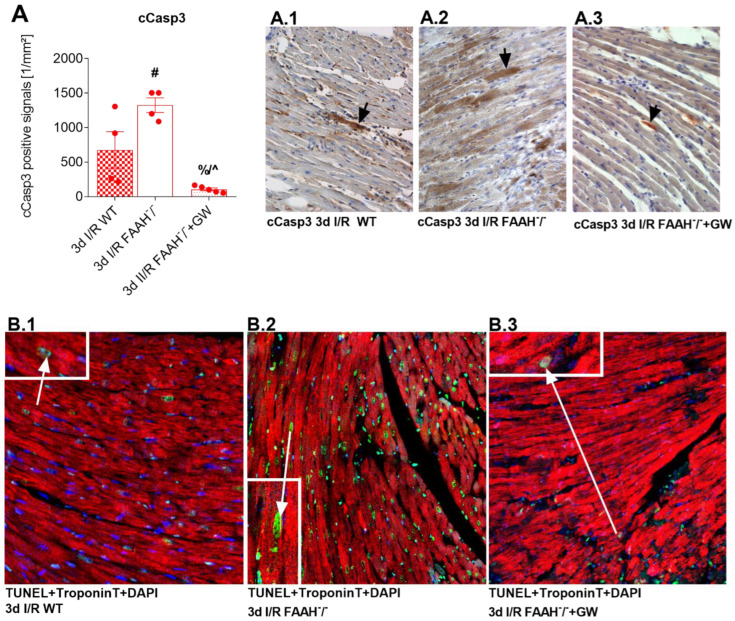
PPAR-α antagonists GW6471 decrease apoptosis and cardiomyocyte loss in FAAH^−/−^-mice. (**A**) Significantly more cCasp3 positive apoptotic cardiomyocytes were observed in FAAH^−/−^ compared to WT- and GW6471 treated FAAH^−/−^ animals. (**A1**–**A.3**) Representative stainings showing cCasp3 positive apoptotic cardiomyocytes (arrows). (**B.1**–**B.3**) Representative TUNEL + Troponin T + DAPI staining underline these findings; arrows indicate apoptotic, TUNEL positive cardiomyocytes. Insets with magnified details show cells of interest. Magnification in (**A1**–**A.3**,**B.1**–**B.3**): 400×. Insets: 800×. cCasp3, Cleaved caspase-3; DAPI, 4′,6-diamidino-2-phenylindole; DMSO, Dimethyl sulfoxide; FAAH, fatty acid amide hydrolase; WT, wild-type; #, *p* < 0.05 FAAH^−/−^ vs. WT; %, *p* < 0.05 FAAH^−/−^+GW vs. FAAH^−/−^+DMSO; ^, *p* < 0.05 FAAH^−/−^+DMSO vs. WT.

**Figure 6 ijms-23-12690-f006:**
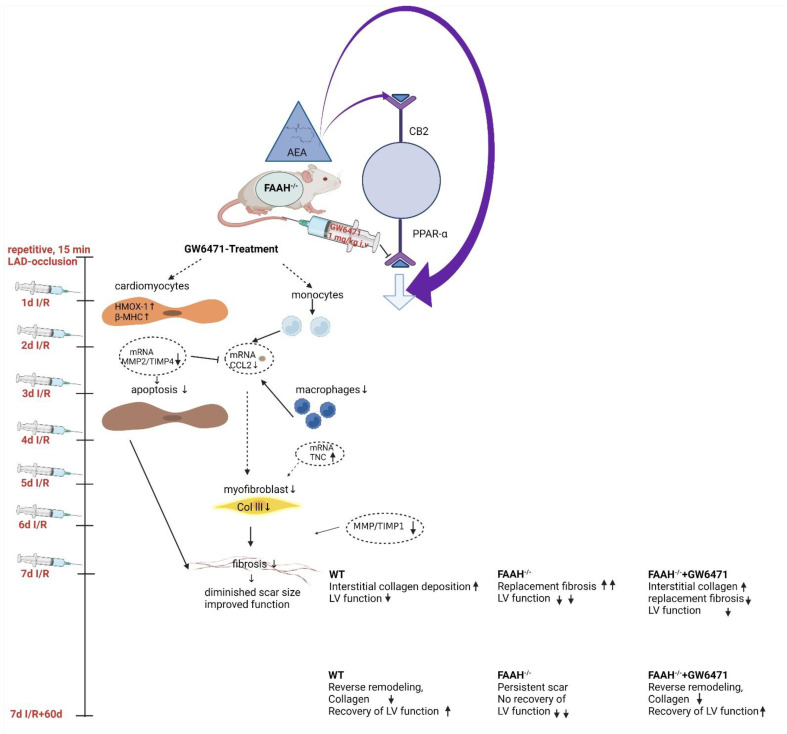
Treatment protocol and schematic overview of the proposed mechanisms of GW6471. Overview scheme of I/R and GW647 treatment and proposed mechanisms of GW6471 when treated 30 min before LAD occlusion. Cardioprotective mechanisms of GW6471 treatment include upregulation of cardiomyocyte adaptive mechanisms to oxidative stress and energy consumption, leading to less cardiomyocyte apoptosis. This is accompanied by modulation of molecular and cellular inflammation, leading to reduced myofibroblast activation, collagen deposition, and thus a smaller scar. AEA, anandamide; CB2, cannabinoid type 2 receptor; CCL2, chemokine; FAAH, fatty acid amide hydrolase; HMOX 1, heme oxygenase 1; IL-10, interleukin-10; LAD, left descending coronary artery; MHC, Myosin heavy chain; MMPs, metalloproteinases; PPAR-α, Peroxisome Proliferator-Activated Receptor; ROS, reactive oxygen species; TIMPs, inhibitors of metalloproteinases; TNC, remodeling-related mediator tenascin-C; WT, wild-type.

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
