# Peer review of "Fatty Acid Amide Hydrolase Deficiency Is Associated with Deleterious Cardiac Effects after Myocardial Ischemia and Reperfusion in Mice"

_ijms, 2022, doi:10.3390/ijms232012690_

Round 1
Reviewer 1 Report
The manuscript is of interest and the experiments address the topic in an extensive way. However major revision are needed.
· The points raised and discussed in the manuscript are often difficult to understand as the sentences are complicated. We suggest a revision of the whole paper in order to simplify and clarify better the concepts expressed. See line 28-29 and 39-42 in the abstract as examples.
· The language should be completely revised (line 195 “adverse”; line 222 “expanded”….)
· The graphs do not show clearly the statistical significance: it would be clearer to indicate the samples that are compared, using square brackets, inserting the asterisks directly above the brackets on the graphs (instead of indicating everything in the figure legends). This should be done in all the figures. The asterisks should be larger.
· Figure 1 and all the other microscopy images: the legends that are reported on the images are not easily readable as they are too small.
· In all graph legends, the way in which the samples are indicated should be the same as in the manuscript and in each figure (example: “DMSO 3d IR” or “3d IR DMSO”; IR and I/R; ß and b). They should report in a clearer way the techniques that are applied.
· Check that all the acronyms are clearly indicated the first time they are used (OPN, ASMAC…)
· The authors should explain the significance of the ratio between MMP/TIMP and the different combinations
· Figure 5: the cleaved Caspase-3 staining is not convincing as it seems to be outside the cells. Better images should be provided. In the TUNEL staining inset with enlargement should be shown to convince that the staining corresponds to nuclei.
· The oligonucleotides sequences are missing.
· In the Flow cytometry material and methods paragraph the antibodies used are not clearly indicated
· The TUNEL protocol is missing
Reviewer 2 Report
This study reports on the phenotype of Fatty acid amide hydrolase (FAAH) knockout mice upon repeated cardiac ischemia/reperfusion injury. Authors observe that loss of FAAH has detrimental effects on cardiac function upon I/R injury as assessed by impaired contractile function, increased fibrosis, and apoptosis in FAAH KO animals compared to WTs. The authors report that these effects can be largely ameliorated by administering the PPAR inhibitor GW6471. Below are my major concerns and specific comments.
Major concerns
- GW6471 is not a selective PPARalpha inhibitor [PMID: 34984327]
- Does activation of PPARalpha exacerbate cardiac damage in mice?
- Provide data that PPARalpha signaling is indeed activated in FAAH KO mice.
- Throughout the manuscript, data on experiments with antagonists should include the WT group, to prove that PPARalpha activation is specific for FAAH KO animals.
- Where possible longitudinal echocardiographic studies should be analyzed by comparing pre-and post-intervention values to control for individual variation, to prove that indeed recovery of cardiac contractility takes place.
- Detrimental effects of FAAH loss in cardiac stress have been reported [PMID: 21070851].
Specific comments
- Check the abstract for clarity.
- Describe in detail at what timepoints GW6471 was administered.
- To prove CM apoptosis, authors should perform containing with CM nucleus specific marker, like PCM1 or NKX2.5.
- Data in Figure 2D contradicts the statement: “FAAH-/--mice … have a 15-fold increase of AEA levels.”
- Authors should assess the classical cardiac stress markers, such as mRNA levels of BNP and ANP in studied mouse models.
- Describe and/or provide references on the origin of FAAH knockout mice.
- Provide evidence that single bolus administration of GW6471 is sufficient to achieve PPAR inhibition. Recent pharmacokinetic data on mice show that the compound is rapidly metabolized with a degradation half-life of 3 min [PMID: 34984327].
- Authors report using diaminobenzidine substrate in IHC, which provides brown staining, however, presented IHC images have a red signal.
- Provide scatter plots for flow Cytometry analyses.
- Mac-2 and ASMAC stainings are substandard and not suitable for quantification.
Reviewer 3 Report
Interesting article on the role of the endocannabinoid system in myocardial ischemia reperfusion. The objectives are clear, authors use appropriate methodology and the authors demonstrate a deep knowledge of the subject.
Minor comments
1. The title is too long
2. Introduction. The authors mention that endocannavinoids play a tremendous role in ischemic cardiomyopathy. This role should be better explained.
Introduction: define FAAH
Discussion: it is too long in some points speculative. Please shorten and moderate some of the statements.
It would be good if in the discussion the authors provide more information about the practical potential of their results.
Round 2
Reviewer 2 Report
1.GW6471 is not a selective PPARalpha inhibitor [PMID: 34984327].
Concern addressed.
2.Does activation of PPARalpha exacerbate cardiac damage in mice
Concern not addressed; see Concern nr.: 3.
3. Provide data that PPARalpha signaling is indeed activated in FAAH KO mice.
The concern is not addressed. Figures 2B and 2C show no difference in PARPalpha target transcript levels between WT and FAAH KO animals treated with DMSO. Indicative of no activation of PPARalpha signaling in FAAH KO animals.
4.Throughout the manuscript, data on experiments with antagonists should include the WT group, to prove that PPARalpha activation is specific for FAAH KO animals.
Concern not addressed. "Reduction" in tripple R concept calls to: "use on the smallest number of animals required to obtain valid information". Without this control, no evidence provided proves that.: PPARalpha activation is specific for FAAH KO animals.
5.Where possible longitudinal echocardiographic studies should be analyzed by comparing pre-and post-intervention values to control for individual variation, to prove that indeed recovery of cardiac contractility takes place.
Concern partially addressed. Where possible, authors should illustrate recovery of cardiac contractility by connecting individual measurements by line and performing statistical analysis appropriate for repeated measures.
6.Detrimental effects of FAAH loss in cardiac stress have been reported [PMID: 21070851].
Concern addressed
Specific comments
- Check the abstract for clarity
Concern addressed.
2. Describe in detail at what timepoints GW6471 was administered.
Concern addressed
3. To prove CM apoptosis, authors should perform containing with CM nucleus specific marker, like PCM1 or NKX2.5.
Concern partially addressed. The authors do not describe how non-CM were excluded from quantification. Without staining for the CM border is not possible to identify the midbody position.
4. Data in Figure 2D contradicts the statement: “FAAH-/--mice ... have a 15-fold increase of AEA levels
Concern addressed.
5. Authors should assess the classical cardiac stress markers, such as mRNA levels of BNP and ANP in studied mouse models
Concern not addressed. BNP and ANP in both experimental and clinical settings are makers for cardiac stress, not hypertrophy.
6. Describe and/or provide references on the origin of FAAH knockout mice.
Concern partially addressed. So were mice in C57BL/6J or 129SvJC57BLy6 background? What was the crossing scheme used to obtain homozygous KO?
7. Provide evidence that single bolus administration of GW6471 is sufficient to achieve PPAR inhibition. Recent pharmacokinetic data on mice show that the compound is rapidly metabolized with a degradation half-life of 3 min [PMID: 34984327].
Concern was partially addressed, but no evidence was provided that used dosing scheme results in efficient PPAR inhibition.
8. Authors report using diaminobenzidine substrate in IHC, which provides brown staining, however, presented IHC images have a red signal
Concern partially addressed, still not clear why aSMAC signal is dark red.
9. Provide scatter plots for flow Cytometry analyses.
Concern addressed
10.Mac-2 and ASMAC stainings are substandard and not suitable for quantification. The reviewer agrees that staining is quantifiable now that correct images are provided for MAC-2 staining.
Round 3
Reviewer 2 Report
The authors have addressed raised concerns.